# The intracellular pathogen *Francisella tularensis* escapes from adaptive immunity by metabolic adaptation

Kensuke Shibata[1,2,3,*] ⓘ, Takashi Shimizu[4,*] ⓘ, Mashio Nakahara[1], Emi Ito[2], Francois Legoux[5], Shotaro Fujii[1], Yuka Yamada[1], Makoto Furutani-Seiki[6], Olivier Lantz[7] ⓘ, Sho Yamasaki[2,8,9,10], Masahisa Watarai[4], Mutsunori Shirai[1]

**Intracellular pathogens lose many metabolic genes during their evolution from free-living bacteria, but the pathogenic consequences of their altered metabolic programs on host immunity are poorly understood. Here, we show that a pathogenic strain of *Francisella tularensis subsp. tularensis* (FT) has five amino acid substitutions in RibD, a converting enzyme of the riboflavin synthetic pathway responsible for generating metabolites recognized by mucosal-associated invariant T (MAIT) cells. Metabolites from a free-living strain, *F. tularensis subsp. novicida* (FN), activated MAIT cells in a T-cell receptor (TCR)–dependent manner, whereas introduction of FT-type *ribD* to the free-living strain was sufficient to attenuate this activation in both human and mouse MAIT cells. Intranasal infection in mice showed that the *ribD*[FT]-expressing FN strain induced impaired Th1-type MAIT cell expansion and resulted in reduced bacterial clearance and worsened survival compared with the wild-type free-living strain FN. These results demonstrate that *F. tularensis* can acquire immune evasion capacity by alteration of metabolic programs during evolution.**

## Introduction

Intracellular pathogens descended from free-living bacteria have adapted to the host environment via evolution of their functional capacities over time (Casadevall, 2008). Such functional alterations in intracellular pathogens arise primarily from gene loss and genetic mutations, through a phenomenon known as "genome reduction." As many of the deleted genes in pathogens encode metabolic pathways, it has been postulated that the associated metabolic changes in the pathogenic strain occur in response to

environmental changes encountered in the host such as availability of nutrients and distinct physiological conditions including changes in pH, oxygen concentration, and osmotic pressures. In addition, acquisition of gene segments by lateral gene transfer can endow bacteria with the capacity to produce virulence factors or other molecules to evade host immune system and promote their survival (Fraser-Liggett, 2005). Therefore, comparative analysis of genomic differences between pathogenic and free-living strains can provide us with valuable insights into the genetic determinants of pathogenesis.

*Francisella tularensis* is a facultative, zoonotic intracellular bacterium that can cause fatal tularemia. *F. tularensis subsp. tularensis* (FT) is the most virulent pathogenic strain and is categorized as a Tier 1 bioterrorism agent by the Centers for Diseases Control and Prevention (Dennis et al, 2001; Dai et al, 2010). In contrast, *F. tularensis subsp. novicida* (FN) is a model free-living strain that rarely causes illness in humans (Birdsell et al, 2009). Although the pathogenic strain FT has unique insertion sequences and genomic rearrangements, FT and FN share a common genome backbone with more than 97% nucleotide identity, supporting emergence from a common ancestor (Larsson et al, 2009). Comparative studies have identified pathogenesis-related genes in FT involved in intracellular replication and virulence (Larsson et al, 2009; Dai et al, 2010; Case et al, 2014; Kingry & Petersen. 2014). These genetic changes result in altered expression of both proteins and metabolites (Larsson et al, 2005; Meibom & Charbit, 2010); however, the consequences of bacterial metabolic changes on host immunity are not well understood.

Mucosal-associated invariant T (MAIT) cells are innate T lymphocytes that recognize microbial metabolites presented by a monomorphic antigen-presenting molecule, major histocompatibility complex class I–like–related molecule 1 (MR1) (Corbett et al, 2014). The most potent metabolic antigen recognized by MAIT cells is 5-(2-oxopropylideneamino)-6-D-ribitylaminouracil (5-OP-RU),

---

[1]Department of Microbiology and Immunology, Graduate School of Medicine, Yamaguchi University, Yamaguchi, Japan  [2]Department of Molecular Immunology, Research Institute for Microbial Diseases, Osaka University, Osaka, Japan  [3]Department of Ophthalmology, Department of Ocular Pathology and Imaging Science, Graduate School of Medical Sciences, Kyushu University, Fukuoka, Japan  [4]Joint Faculty of Veterinary Medicine, Laboratory of Veterinary Public Health, Yamaguchi University, Yamaguchi, Japan  [5]INSERM U932, PSL University, Institut Curie, Paris, France  [6]Systems Biochemistry in Pathology and Regeneration, Graduate School of Medicine, Yamaguchi University, Ube, Japan  [7]INSERM U932, PSL University, Laboratoire d'Immunologie Clinique, Centre d'Investigation Clinique en Biothérapie, Institut Curie (CIC-BT1428), Paris, France  [8]Laboratory of Molecular Immunology, Immunology Frontier Research Center, Osaka University, Osaka, Japan  [9]Division of Molecular Design, Medical Institute of Bioregulation, Kyushu University, Fukuoka, Japan  [10]Division of Molecular Immunology, Medical Mycology Research Center, Chiba University, Chiba, Japan

Correspondence: kshibata@yamaguchi-u.ac.jp
*Kensuke Shibata and Takashi Shimizu contributed equally to this work.

which is generated from an intermediary molecule 5-amino-6-D-ribitylaminouracil (5-A-RU) as part of the riboflavin synthetic pathway, present in microbes such as bacteria and fungi but not in mammals (Fig 2A). The riboflavin synthetic pathway consists of several *rib* genes that together work to convert GTP to riboflavin (Bacher et al, 2000). The conversion step resulting in generation of 5-A-RU depends on the enzyme RibD, and consequently, *ribD* deficiency abrogates the ability of microbes including *Salmonella epidermidis*, *Escherichia coli*, and *Lactococcus lactis* to produce 5-A-RU and activate MAIT cells (Corbett et al, 2014; Soudais et al, 2015; Constantinides et al, 2019).

MAIT cells have been shown to play a role in host immunity to *F. tularensis*. When MAIT cells were co-cultured with macrophages infected with the avirulent *F. tularensis* live vaccine strain (LVS), MAIT cells produced inflammatory mediators such as IFN-γ, TNF-α, IL-17A, nitric oxide, and granulocyte macrophage colony-stimulating Factor (GM-CSF) in a TCR-dependent manner (Meierovics et al, 2013; Meierovics & Cowley, 2016). After infection with the LVS, MAIT cell deficiency impaired bacterial clearance (Meierovics et al, 2013), and MAIT cell transfer experiment showed that IFN-γ produced by MAIT cells contributes to the protective response (Zhao et al, 2021) by facilitating accumulation of DCs at the site of infection (Meierovics & Cowley, 2016). These results suggest that the avirulent *Francisella* strain induces protective responses through TCR-dependent MAIT cell activation. However, it remains to be determined whether MAIT cells also have protective roles against the pathogenic strain FT.

In the present study, we used a comparative transcriptomics approach to define putative mutations contributing to immune recognition differences between pathogenic and avirulent strains of *F. tularensis*. We identified five amino acid substitutions in RibD in the pathogenic strain FT compared with the free-living strain FN. The activation of MAIT cells was attenuated by introduction of the RibD substitutions into the free-living strain FN. In agreement with these in vitro effects, infection of mice with the *ribD*[FT]-expressing FN (FNribD[FT]) strain led to reduced MAIT cell expansion compared with infection with the parental strain. Moreover, these substitutions contributed to bacterial pathogenesis as mice infected with the FNribD[FT] strain had higher bacterial loads in the lung and shortened survival. These results suggest that, during adaptation, altered RibD function has been selected in the pathogenic strain to escape recognition by adaptive immunity.

## Results

### The pathogenic *Francisella* strain has unique metabolic programs

As genetic changes are the canonical drivers for acquisition of virulence by microbes, extensive efforts have been made to apply comparative genome sequencing analysis to pathogenic and free-living *Francisella* strains (Rohmer et al, 2007; Larsson et al, 2009). To better understand gene expression differences between these strains, we performed RNA sequencing analysis. We identified 148 genes whose expression was not detected in the pathogenic strain FT. Kyoto Encyclopedia of Genes and Genomes (KEGG) pathway analysis revealed that these genes were highly enriched in metabolic pathways (Fig 1A). For example, transcripts

of genes involved in amino acid biosynthesis such as *lysC* (lysine); *leuB*, *leuC*, and *leuD* (valine and leucine); and *tyrA* (tyrosine) were not detected in the pathogenic strain FT (Fig 1B), consistent with previous studies (Rohmer et al, 2007; Meibom & Charbit, 2010). In addition, we detected no expression of genes such as *pip*, *dapA*, *dapB*, *dapD*, *aspC*, *asnB*, and *alr*, which are involved in the synthesis of other amino acids, in the pathogenic strain FT (Fig 1B).

Further analysis showed that 352 genes had decreased expression in the pathogenic strain FT compared with the free-living strain FN (Fig 1C). These included genes such as *appB* and *appC*, which are related to energy consumption (Fig 1C and D) as shown in a previous study (Casadevall, 2008). We also found down-regulation of many genes related to amino acid metabolism including *ilvN*, *ilvB*, and *ilvC*, which are responsible for valine synthesis (Fig 1C and D). The attenuated expression of *ilvB* and *ilvC* would be consistent with previously reported partial deletions or nonsense mutations in these genes in *F. tularensis* (Rohmer et al, 2007). In contrast, in the pathogenic strain FT, we observed significant up-regulation of genes involved in biotin (vitamin B7) synthesis (*bioA*, *bioB*, *bioC*, *bioD*, and *bioF*) which is known to be important for phagosomal escape (Fig 1C and E) (Napier et al, 2012). Furthermore, the *aroA*, *aroB*, and *aroC* genes responsible for phenylalanine, tyrosine, and tryptophan biosynthesis were also up-regulated, and *aroC* has been shown to be a virulence factor used by the intracellular pathogen *Salmonella enterica serovar Typhimurium* (*Salmonella typhimurium*) (Hone et al, 1991) (Fig 1C and E). These results suggest that the pathogenic strain of *F. tularensis* is adapted to the host environment by acquiring unique metabolic regulation when compared with its free-living relative. Thus, we referred to this phenomenon as metabolic adaptation in this study.

### The pathogenic *Francisella* strain fails to activate MAIT cells

MAIT cells recognizing microbial metabolite antigens generated during riboflavin synthesis are an established part of the host immune response to avirulent *F. tularensis* (Meierovics et al, 2013; Corbett et al, 2014; Meierovics & Cowley, 2016; Zhao et al, 2021). To examine whether the metabolic gene expression changes in the pathogenic strain have an impact on MAIT cell activation, expression profiles of riboflavin synthesis-related genes such as *ribAB*, *ribD*, *ribH*, and *ribC* involved in generation of a MAIT cell antigen were compared between the pathogenic and free-living strains (Fig 2A). Despite comparable expression levels of these genes (Fig 2B), we detected five amino acid substitutions in RibD including I59T, V61A, G62V, H80C, and Q254R in the pathogenic strain (Fig 2C). Four of the substitutions are located in the zinc-binding region of the cytidine and deoxycytidylate deaminase domain, and one substitution is in the bacterial bifunctional deaminase-reductase domain (InterPro database, https://www.ebi.ac.uk/interpro/) (Fig 2C). No substitutions were found in RibAB, RibH, and RibC. Among the five substitutions in RibD, H80C and Q254R were shared with seven other virulent strains, based on genome sequences deposited in the KEGG database (Fig 2D). Given that *ribD* is crucial for generating a MAIT cell antigen 5-OP-RU in other bacterial strains (Corbett et al, 2014; Soudais et al, 2015; Constantinides et al,

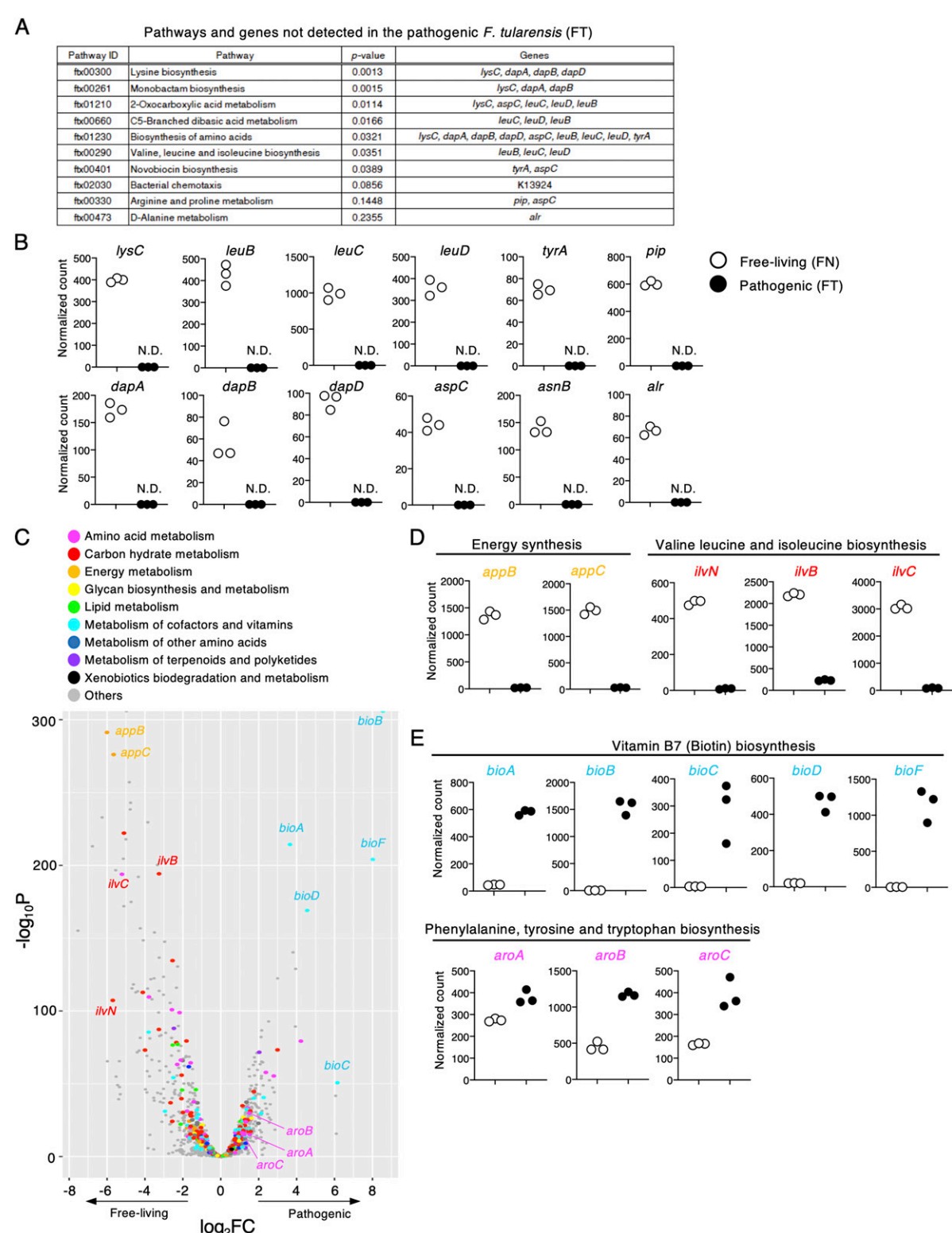

**Figure 1. FT has unique metabolic programs.**

RNA sequencing of free-living and pathogenic *F. tularensis* strains. **(A)** The top 10 pathways expressed in the free-living (FN) but not the pathogenic strain (FT) were calculated by Fisher's exact test. A total of 148 genes were detected only in the free-living strain and were used for this analysis. **(A, B)** Plots of normalized counts of each gene noted in (A) in both strains. **(C)** Volcano plot analysis of 1,229 genes detected in both the pathogenic and free-living strain. X- and y-axis show the $\log_2$FC and $\log_{10}P$-values, respectively. **(D, E)** Plots of the normalized read counts for genes down-regulated (D) or up-regulated (E) in the pathogenic strain.

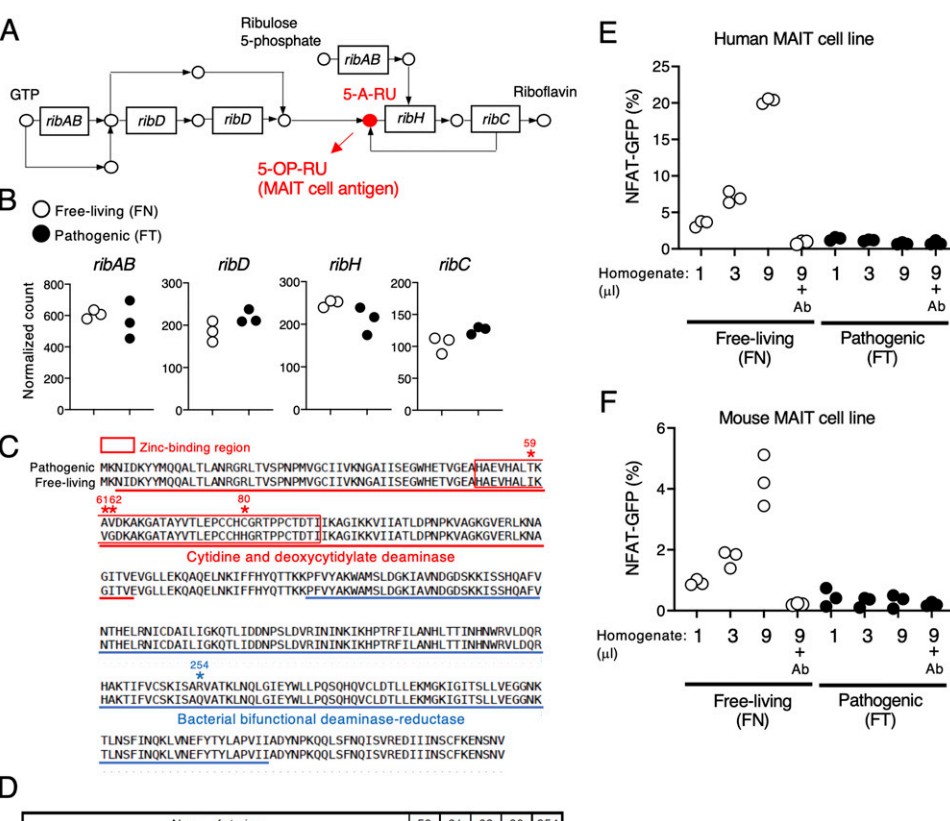

**Figure 2. FT has mutations in *ribD* and does not activate MAIT cells.**
**(A)** Schematic of the riboflavin synthetic pathway. The precursor of the MAIT cell antigen presented by MR1 is highlighted in red. **(B)** Plots of normalized read counts of genes *ribAB*, *ribD*, *ribH*, and *ribC* in the free-living and pathogenic strains.
**(C)** Comparison of the amino acid sequence of RibD in the pathogenic and free-living strain. Boxed amino acids (red) are the zinc-binding region of the cytidine and deoxycytidylate deaminase domain. Underlined amino acid sequences are catalytic domains. **(D)** Summary table of amino acids at five defined positions in RibD in free-living (black) and pathogenic (red) strains as found in the KEGG database.
**(E, F)** GFP-reporter activities of cells expressing human (E) and mouse (F) MAIT TCRs after stimulation with escalating amounts (1, 3, 9 µl) of total metabolites from the indicated strains that were cultured in the presence of human and mouse MR1-expressing cells, with or without anti-MR1 antibody (Ab) (10 µg/ml). Data are representative of three independent experiments.

2019), we hypothesized that these substitutions in the pathogenic *Francisella* strain may have a role on MAIT cell activation. To test this, we prepared bacterial metabolites from equal cell numbers of FT and FN bacteria and compared their agonistic activity using human and mouse MAIT TCR-expressing reporter cells that induce GFP expression in response to TCR-dependent recognition of their cognate antigens (Fig S1A). Indeed, the MAIT TCR-expressing reporter cells were activated after stimulation of an agonist 5-OP-RU, which was completely blocked by addition of an antagonist Ac-6-FP (Fig S1B). Notably, although the bacterial metabolites from the free-living strain induced a dose-dependent increase in reporter fluorescence, this activity was not induced in either the human and murine MAIT reporter cell lines in response to the pathogenic FT strain (Figs 2E and F and S2A and B). Importantly, the agonistic activity induced by metabolites from the free-living FN strain was completely abolished by addition of an anti-MR1 blocking antibody (Fig 2E and F). These results suggest that metabolites generated by the pathogenic strain FT have lost their ability to activate MAIT cells via the TCR.

## RibD is essential for activation of MAIT cells

To directly test whether *ribD* in *F. tularensis* is involved in the activation of MAIT cells, the *ribD* gene in the free-living strain was deleted by homologous recombination (FNΔ*ribD*). The *ribD* gene was not essential for cell growth as the optical density of the FNΔ*ribD* strain was identical to the parental strain over a culture time course (Fig 3A). We confirmed that the *ribD* gene was selectively deleted in the FNΔ*ribD* strain using RNA sequencing (Fig 3B). Metabolites from the FNΔ*ribD* strain were unable to activate both human and murine MAIT reporter cell lines (Fig 3C and D), supporting an essential role for *ribD* in MAIT cell activation similar to other bacteria (Corbett et al, 2014; Soudais et al, 2015; Constantinides et al, 2019). To specifically test the importance of the amino substitutions in FT RibD, these changes were introduced into the free-living strain. As with the FNΔ*ribD* strain, the introduction of these mutations did not perturb bacterial growth (Fig 3A). However, the ability of bacterial metabolites to activate MAIT cells was decreased in the FNribD^FT strain compared with the parental strain

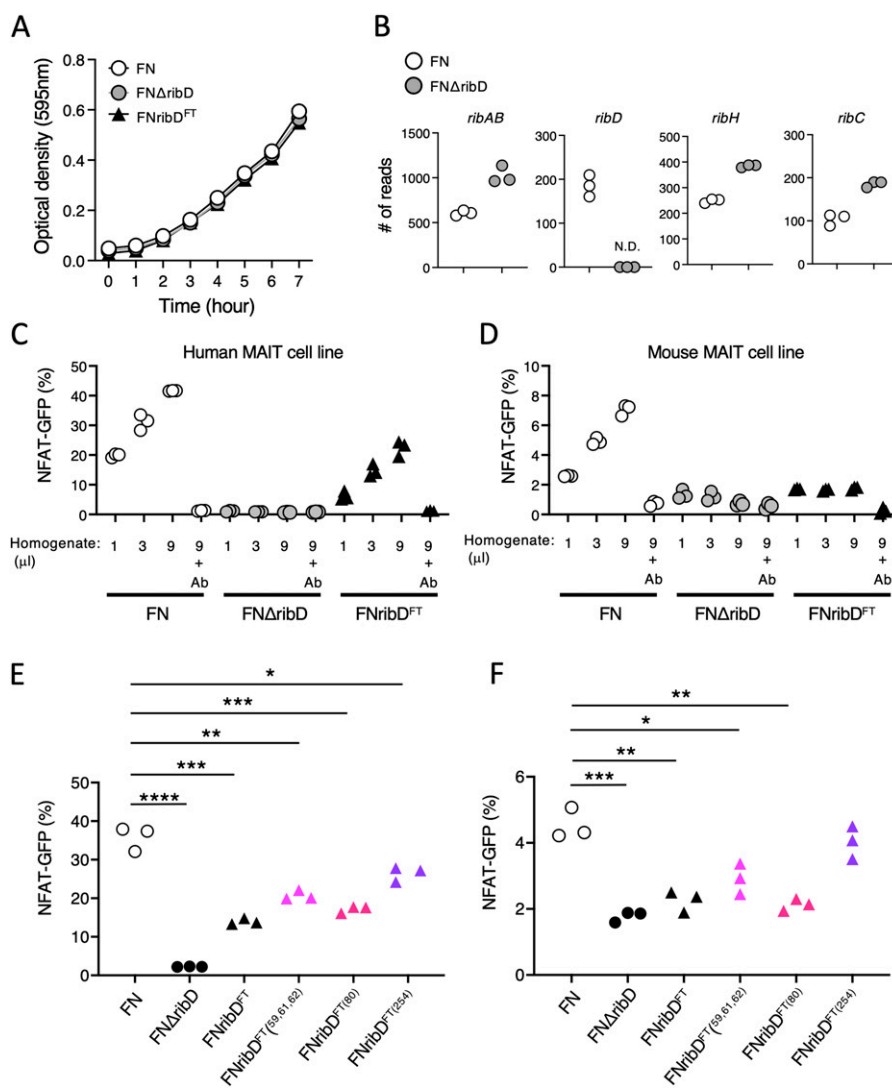

**Figure 3. RibD is required for MAIT cell activation.**
**(A)** Growth curve of the indicated strains after culture with shaking shows no difference in change in optical density over time. **(B)** Plots show normalized read counts of genes in the riboflavin synthetic pathway in free-living parental FN or FNΔ*ribD* strains. **(C, D, E, F)** GFP-reporter activities of cells expressing human (C, E) and mouse (D, F) MAIT TCR after stimulation with escalating amounts (1, 3, 9 µl) (C, D) or 9 µl (E, F) of total metabolites from FN, FNΔ*ribD*, FNribD[FT], FNribD[FT (56, 61, 62)], FNribD[FT (80)], and FNribD[FT (254)] strains cultured in the presence of human or mouse MR1-expressing cells. *$P < 0.05$, **$P < 0.01$, ***$P < 0.001$ by one-way ANOVA, followed by Dunnett's multiple comparison test. Data are representative of three independent experiments.

(Figs 3C and D and S2C and D). To further dissect which amino acids are required for the MAIT cell activation, a series of individual mutant strains was generated (FNribD[FT(59,61,62)], FNribD[FT(80)], FNribD[FT(254)]). Among amino acid substitutions at the position 80 and 254 conserved in the pathogenic strain FT (Fig 2D), H80C substitution attenuated both human and mouse MAIT TCR stimulation (Figs 3E and F and S2C and D). Introduction of the other amino acid substitutions such as I59T, V61A, G62V, and Q254R could also reduce this capacity, albeit to a lesser extent than H80C alone (Figs 3E and F and S2C and D). These results together indicate that the RibD amino acid substitutions in the pathogenic strain FT are sufficient to attenuate MAIT cell activation, likely because of a reduction in MAIT cell antigen production.

## The FNribD[FT] strain enhances virulence of *F. tularensis*

To determine the pathological function of the amino acid substitutions in RibD, we employed a mouse model of intranasal FN infection. Mice inoculated with the FNribD[FT] strain succumbed to

accelerated lethality compared with those infected with the parental strain (Fig 4A). An in vitro infection experiment showed that bacterial translocation to the intracellular compartment was not perturbed by the absence or mutation of *ribD* (Fig 4B). Indeed, in the infected lung, both FN and FNribD[FT] bacteria were present (Fig 4C); however, mice inoculated with the FNribD[FT] strain showed more severe histology including pulmonary airway hemorrhage and complete loss of normal alveolar architecture (Fig 4D). Bacterial clearance in the lung and spleen was impaired upon infection with the FNribD[FT] strain (Fig 4E). The exacerbated pathogenic responses and increased bacterial burdens in mice inoculated with the FNribD[FT] strain were associated with reduced numbers of MAIT cells in the lung (Fig 4F). This in vivo MAIT cell expansion was completely dependent on *ribD* as mice infected with the FNΔ*ribD* strain showed no MAIT cell expansion (Figs 4G and H and S3) and a corresponding decrease in the accumulation of DCs and macrophages in the lung (Figs 4H and S3). After infection with the parental strain, the expanded MAIT cells acquired the capacity to produce IFN-γ (Fig 4I and J) which is essential for protection against *Francisella* infection

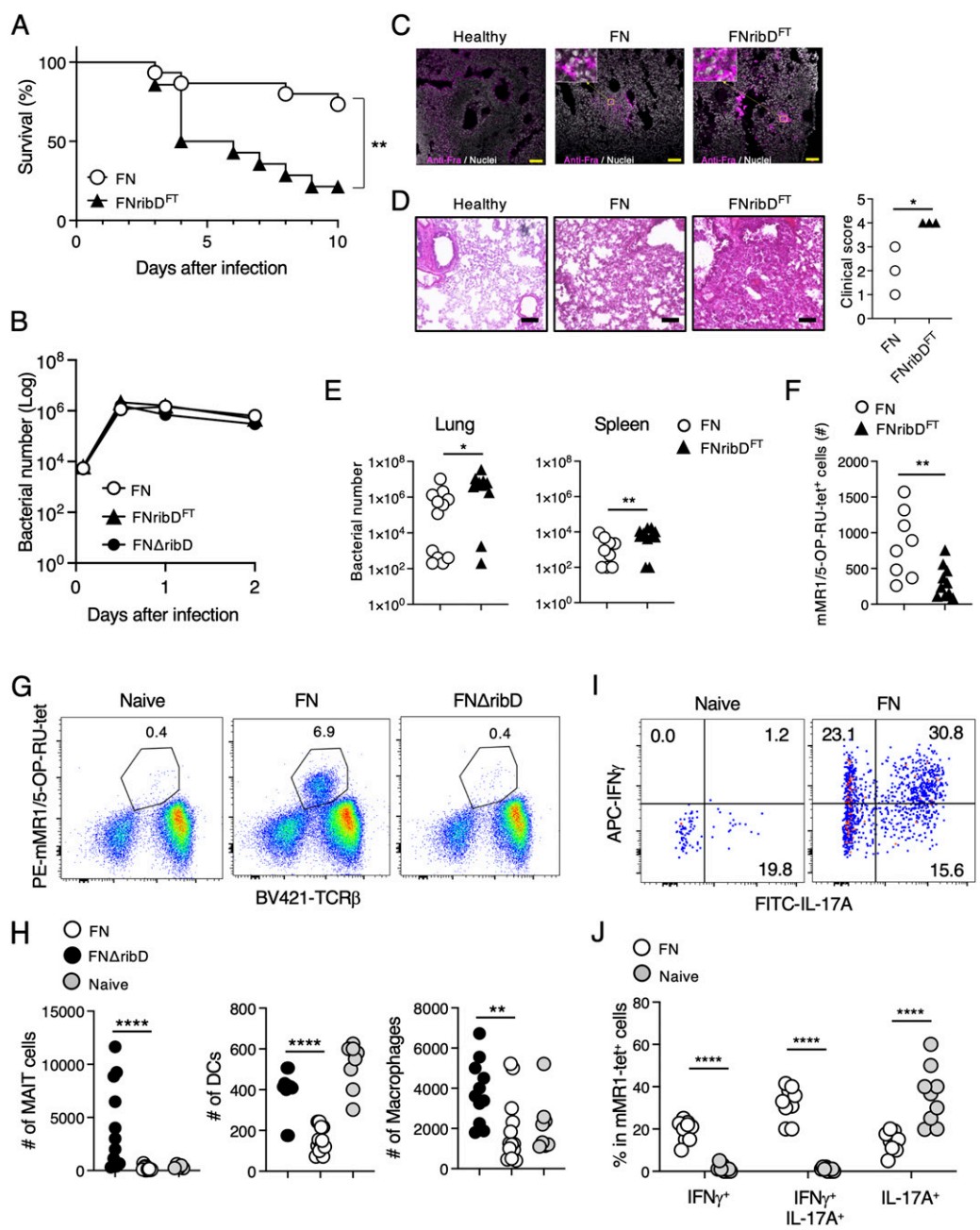

**Figure 4. Amino acid substitutions in RibD increase the pathogenesis of *F. tularensis*.**
**(A)** Survival rates of mice after intranasal infection with the FNribD^FT or its parental strain FN (n = 15 in each group). Asterisks indicate statistical significance determined by logrank tests (**P < 0.01). **(B)** Graph shows the changes of bacterial numbers after infection in vitro. After infection with indicated strains to THP-1 cells, bacteria present in the cells were calculated at each time point. **(C, D)** Histological analysis of inflammation in the lung by immunohistochemistry (C) or H&E staining with quantified data (D) 4 d after intranasal infection. **(C, D)** Scale bar shows 10 μm (C) or 50 μm (D), respectively. Statistical significance was determined by unpaired two-tailed *t* tests (*P < 0.05).
**(E)** Bacterial burdens in the indicated tissues 4 d after infection (n = 14 in each group). Statistical significance was determined by unpaired two-tailed *t* tests (*P < 0.05, **P < 0.01).
**(F)** Numbers of mMR1/5-OP-RU-tet^+ MAIT cells in the lung 4 d after infection. Statistical significance was determined by unpaired two-tailed *t* tests (**P < 0.01). **(G, H)** Flow cytometric analysis of lung 14 d after infection with the indicated strain. **(G)** mMR1/5-OP-RU-tet^+ MAIT cells in the lung are shown after gating on total lymphocytes. **(H)** Plots show the absolute numbers of mMR1/5OP-RU-tet^+ MAIT cells, mCD11c^+ DCs, and mF4/80^+ macrophages. Statistical significance was determined by one-way ANOVA, followed by Dunnett's multiple comparison test (**P < 0.01, ****P < 0.0001). **(I, J)** Flow cytometric analysis of lung MAIT cells 9 d after infection with the avirulent strain FN. **(I)** After gating on mMR1/5OP-RU-tet^+ cells, IFNγ and IL-17A production was analyzed by intracellular staining. **(J)** Graph shows frequencies of the indicated populations before and after infection with FN. Statistical significance was determined by unpaired two-tailed *t* test (****P < 0.0001). Data in (A, E, F, H, J) are combined from three independent experiments. Data in (B, C, D, G, I) are representative of three independent experiments.

(Zhao et al, 2021). In agreement with the previous study (Rahimpour et al, 2015), resident MAIT cells in the lung were IL-17A–producing cells (Fig 4I and J), suggesting that functional differentiation and/or expansion of Th1-type MAIT cells occurs after infection with the avirulent *Francisella* strain (Zhao et al, 2021). These results suggest that specific ribD amino acid substitutions from the pathogenic strain are sufficient to attenuate Th1-type MAIT cell expansion and increase the pathogenesis of free-living, avirulent *F. tularensis*.

## Discussion

Metabolic adaptation is a frequent occurrence in bacterial pathogens upon the transition from the free-living state to an intracellular environment (Moran, 2002). In the present study, we found that amino acid substitutions in a component in the riboflavin synthetic pathway, *ribD*, of the intracellular pathogen *F. tularensis* contribute to escape of the pathogen from recognition by T cells. This finding suggests that metabolic changes in pathogenic bacteria serve purposes beyond adaptation to the environment and furthermore allow pathogens to survive without host detection. This can be directly tested by introducing WT *ribD* to the FT strain for the future study. As *ribD* is present in a wide variety of pathogens (Vitreschak et al, 2002) and is essential for MAIT cell activation, it will be interesting to determine whether *ribD*-mediated metabolic adaptation is also involved in the pathogenesis of other bacteria. Indeed, multidrug-resistant *S. typhimurium* reduces MAIT cell activation property by altered gene expressions in the riboflavin synthetic pathway (Preciado-Llanes et al, 2020). Augmentation of riboflavin synthetic pathway in *Mycobacterium tuberculosis* enhances MAIT cell–dependent protective responses (Dey et al, 2022). Intriguingly, some clinical isolates of multidrug resistant *M. tuberculosis* and *Mycobacterium. leprae* acquire *ribD* mutations (Zhang et al, 2015; Benjak et al, 2018), although their impact on the generation of MAIT cell antigens and the host immune response remain to be determined.

Despite the paradigm of genome reduction in pathogens leading to loss of metabolic pathway-related genes, we additionally found up-regulated genes in the pathogenic *Francisella* strain to be involved in biosynthesis of vitamin B7 (biotin) and other aromatic metabolites. Biotin has been shown to be required for escape from the phagosome to the cytosol (Napier et al, 2012), and *aroC*, which is responsible for aromatic amino acid synthesis in the pathogenic *Francisella* strain, contributes to its intracellular growth capacity (Cunningham et al, 2020). These results suggest that combined use of selective inhibitors of pathogenesis pathways together with MAIT cell antigens would be an alternative strategy for more effective vaccine development.

The immune system discriminates pathogens from the self by recognizing microbial cellular components such as DNA, RNA, lipids, and proteins, known as pathogen-associated molecular patterns (Medzhitov, 2001; Miyake & Yamasaki, 2012; Gong et al, 2020). Given these characteristics, we propose that MR1 acts as a sensor and presenter of pathogen-associated molecular patterns by binding metabolites including 5-OP-RU generated by intracellular pathogens, which are not present in mammals but widely conserved among microbial species. MR1 is furthermore highly conserved

among mammals (Boudinot et al, 2016), supporting the physiological importance of this metabolite-sensing system during evolution.

Previous studies have shown that IFN-γ- and *Francisella*-specific antibodies are essential for protection against the avirulent *Francisella* strain (Kirimanjeswara et al, 2007). Furthermore, the importance of IFN-γ produced by MAIT cells in the context of *Francisella* infection was also recently reported (Zhao et al, 2021); however, the contribution of MAIT cells to *Francisella*-specific antibody production is not known. It is interesting to determine whether Th1-type MAIT cells provide help to B cells to facilitate the *Francisella*-specific antibody responses, as is the case for infection with other pathogens (Leung et al, 2014; Rahman et al, 2020).

Because whole genomes of pathogenic and free-living strains of *Francisella* have become publicly accessible, many candidate genes related to bacterial pathogenesis have been identified (Casadevall, 2008). Various candidate pathogenesis genes encoding proteins involved in cell wall synthesis, intracellular growth, and antibiotic resistance have been tested in infection mouse models (Marohn & Barry, 2013). However, despite extensive efforts, to understand the pathogenic mechanisms and mediators of *F. tularensis*, no tularemia vaccine has been approved. In the conventional vaccine development, protein-derived peptides presented by polymorphic antigen-presenting molecules that activate conventional T cells have been tested. In contrast, riboflavin precursors presented by monomorphic MR1 lead to MAIT cell activation. Thus, we propose that a metabolite-targeted strategy, rather than traditional protein-targeted approaches, may open a new avenue for vaccine development to protect from infection with pathogenic *Francisella* to a wider range of patients in addition to therapeutic potential of the strategy against cancers and autoimmune diseases (Rouxel et al, 2017; Crowther et al, 2020; Yan et al, 2020; Ruf et al, 2021; Yamana et al, 2022).

## Materials and Methods

### Mice

Four-week-old C57BL/6 mice were purchased from SLC. This study was approved by the Committee of Ethics on Animal Experiments in the Animal Research Committee of Yamaguchi University. Experiments were carried out under the control of the Guidelines for Animal Experiments.

### Cell lines

Human and mouse MAIT αβTCR-expressing reporter cell lines (TCRs were derived from 4L4T, 12F12) were generated by retroviral gene transduction of MAIT αβTCRs to TCR-negative mouse T-cell hybridoma with an NFAT-GFP reporter gene (Matsumoto et al, 2021). Human MAIT αβTCR sequences were from the Protein Data Bank. Murine MAIT αβTCR sequences (12F12) were kindly provided by Prof. Olivier Lantz (Institut Curie) (Tilloy et al, 1999). For antigen-presenting cells, human and mouse MR1-expressing cells were generated by retroviral gene transduction of human and mouse

MR1 to human melanoma cell line A375 and mouse fibroblastic cell line NIH3T3. Human monocyte cell line THP-1 was used for infection experiment in vitro. All cell lines were maintained in in RPMI (Sigma-Aldrich) supplemented with 10% fetal calf serum at 37°C in a $CO_2$ incubator.

## Preparation of bacteria and their total metabolites

The FT strain SCHU P9 (Uda et al, 2014) and the FN strain U112 were cultured in brain heart infusion broth (Becton, Dickinson and Company) supplemented with cysteine (BHIc) (Mc Gann et al, 2010). Bacterial strains were diluted to $OD_{595}$ = 0.001 and cultured with shaking for 12 h. Approximately $2 \times 10^8$ bacterial cells were washed with PBS twice and suspended in 500 $\mu$l of $H_2O$. FN cells were disrupted by sonication using VP-050 (Taitech) at PWR 80 for 10 s × 10 times on ice. FT cells were disrupted using EZ-Beads (Promega) and beads crusher $\mu$T-01 (Taitech) for 10 s × 10 times while being cooled on ice. After centrifugation at 13,000$g$ for 20 min at 4°C, supernatants were filtered through 0.22-$\mu$m filter and were stored at –80°C until use.

## Generation of FN-derived mutants

Table S1 lists the primers used in this study. PCR was performed using KOD-Plus-Neo polymerase (Toyobo), and ligation was performed using an In-Fusion HD Cloning Kit (Takara Bio). Plasmids were used to transform FN by electroporation. Bacterial cells were suspended in 0.5 M sucrose with 2 $\mu$g of plasmid DNA and were electroporated using a Bio-Rad micropulser (Bio-Rad) at 3.0 kV, 10 $\mu$F, and 600 Ω with 0.2-cm cuvette; transformants were precultured in CDM medium overnight. To select the transformed bacteria, the preincubated bacteria were cultured on BHIc agar plates containing 30 $\mu$g/ml kanamycin. A deletion mutant of $ribD$ (FN$\Delta ribD$) was generated via homologous recombination using the $Francisella$ suicide vector pFRSU (Shimizu et al, 2019). The upstream and downstream regions of $ribD$ (1.5 kbp each) were cloned into the BamHI site of pFRSU to generate pFRSU-$\Delta$ribD. The pFRSU-$\Delta$ribD vector (2 $\mu$g) was used to transform FN. Transformants were cultured in BHIc without antibiotics overnight and then plated on BHIc plates containing 5% sucrose. The deletion of the $ribD$ gene was confirmed by PCR. To generate the FNribD$^{FT}$ strain, $ribD$ gene of FT was cloned into pFRSU-$\Delta$ribD (pFRSU-ribD$^{FT}$). To generate FNribD$^{FT(56,61,62)}$, $ribD$ gene of FN was cloned into pFRSU-$\Delta$ribD (pFRSU-ribD), and mutation was induced by PCR (pFRSU-ribD$^{FT(56,61,62)}$). pFRSU-ribD$^{FT(56,61,62)}$ was used to transform FN. To generate FNribD$^{FT(80)}$, mutation was induced in pFRSU-ribD by PCR (pFRSU-ribD$^{FT(80)}$). pFRSU-ribD$^{FT(80)}$ was used to transform FN. To generate FNribD$^{FT(254)}$, mutation was induced in pFRSU-ribD by PCR (pFRSU-ribD$^{FT(254)}$). pFRSU-ribD$^{FT(254)}$ was used to transform FN.

## Infection experiment in vitro

THP-1 cells ($4 \times 10^5$ cells/well) were incubated in a 48-well tissue culture plate with 200 nM PMA for 48 h. After the incubation, indicated strains were added at a MOI = 1 and were centrifuged for 10 min at 300$g$ and incubated for 1 h at 37°C. The cells were washed three times with RPMI 1640 medium, and extracellular bacteria were

killed with gentamicin at 50 $\mu$g/ml for 1 h. The cells were then incubated in fresh medium at 37°C for the times indicated. To measure intracellular growth, the cells were washed with PBS and then lysed with 0.1% Triton X-100 in CDM. The number of CFUs was determined on BHIc agar plates by plating serial dilutions of cultures.

## Infection experiment in vivo

C57BL/6J mice (4-wk-old females; SLC) were intranasally inoculated with 10 $\mu$l of $7.5 \times 10^8$ CFU bacteria under anesthesia with 0.08 $\mu$g medetomidine hydrochloride (Domitor; Orion Diagnostica), 25 $\mu$g midazolam (Dormicum; AstellasPharma), and 0.3 $\mu$g butorphanol tartrate (Vetorphale; Meiji Seika). Mice were euthanized via $CO_2$ treatment for 5 min. Bacterial numbers were quantified by harvesting lungs and spleens from the infected mice. The organs were homogenized in PBS using bio smasher II (Nippi), and the homogenates were cultured on BHIc containing 1.5% agar (Wako Laboratory Chemicals) at 37°C for 2 d.

## RNA sequencing analysis for *F. tularensis*

Free-living and pathogenic *Francisella* strains were cultured to exponential phase at 37°C in a brain heart infusion broth (Becton, Dickinson and Company) supplemented with cysteine. After the culture, total RNA was prepared using NucleoSpin RNA (Takara Bio). After removal of rRNA using Illumina Ribo-Zero Plus rRNA Depletion Kit (Illumina), the TruSeq Stranded mRNA Library Prep Kit (Illumina) was used with 1 $\mu$g of total RNA for the construction of sequencing libraries. Fragment size of the libraries was confirmed with the Agilent 2100 Bioanalyzer (Agilent). Libraries were sequenced on an Illumina NovaSeq 6000 in single-end mode (100-bp reads). The raw reads were mapped onto the NZ_CP009633.1 *F. tularensis subsp. novicida* U112 sequence using RSEM. Comparative expression analysis was performed by edgeR. Raw read counts were normalized based on total read counts. Statistical significance of each gene was calculated as $P$-value and fold-change (| logFC |). For pathway analysis, 148 genes detected only in nonvirulent *F. tularensis* with statistical significance ($P < 0.05$, | logFC | > 1) were analyzed in the KEGG database (https://www.genome.jp/kegg/).

## Flow cytometric analysis

Cells were stained in PBS containing 2% FCS. FITC-conjugated anti-mouse (m) IL-17A (TC11-18H10.1) mAb, PerCP-Cy5.5–conjugated anti-mI-A/I-E (M5/114.15.2) mAb, APC-conjugated anti-mCD3e (2C11), anti-mCD11c (N418), anti-mF4/80 (BM8) and anti-mIFNγ (XMG1.2) mAbs, BV421-conjugated anti-mTCRβ (H57-597) and anti-mF4/80 (BM8) mAbs, BV510-conjugated anti-mI-A/I-E (M5/114.15.2) mAb, and BV570-conjugated anti-mCD45 (30-F11) mAb were purchased from BioLegend. For cell surface staining, cells were incubated for 20 min at 4°C. For tetramer staining, before surface staining, cells were stained with mouse MR1 tetramers (provided by NIH core facility) for 30 min on ice. Dead cells were excluded from analysis by staining with 7-amino-actinomycin D-containing viability staining solution (BioLegend). Flow cytometry was performed after staining

on a Novocyte flow cytometer (ACEA Biosciences). The data were analyzed using FlowJo software version 10.5.0 (BD Biosciences).

## Compounds

Ac-6-FP was purchased from Schircks Laboratories. 5-A-RU was purchased from Toronto Research Chemicals. 5-OP-RU was generated by reacting 5-A-RU with an equivalent molar ratio of methylglyoxal (Sigma-Aldrich).

## Histological analysis

Histological analysis was performed after freshly isolated tissue samples were fixed with formalin and then embedded in paraffin or OCT compound (Sakura Finetek). Sections were stained with hematoxylin and eosin for detecting inflammatory cells. To generate an antiserum against FN, mice were immunized with FN homogenates. OCT compound-embedded sections of mouse pancreatic tissues were stained with an antiserum against FN overnight at 4°C. For detecting nuclei, DAPI (1.7 μg/ml; WAKO) was used. Stained tissues were visualized using a Leica TCS SP8 confocal microscope (Leica). The inflammation of the lungs was assessed in a blinded manner following a protocol as described previously (Doganci et al, 2008). Inflammation was graded using a four-tier score where: 0, no inflammation; 1, rare occasional inflammatory cells around isolated peribronchial blood vessels; 2, accumulations of scant inflammatory cells around peribronchial vessels in more than one site; 3, multifocal inflammation around peribronchial vessels, easily visible at ×4 magnification; and 4, severe peribronchial inflammatory infiltration at multiples sites.

## Statistics

Statistical significance was calculated using Prism software (GraphPad). Differences with values for $P < 0.05$ were considered statistically significant.

# Data Availability

Bulk RNA-sequencing data have been deposited at the DNA Data Bank of Japan (DDBJ) database under the accession number GSE174793.

# Supplementary Information

# Acknowledgments

We thank M Kawano, R Nakamura, and M Shirahase for technical support; M Kazuki (Cell Innovator Co., Ltd., Fukuoka, Japan) for assistance with the sequencing analysis and useful discussion. This work was supported by grants from the Japan Society for the Promotion of Science KAKENHI (JP20K07539, JP19K08424) and Core Clusters for Research Initiatives of Yamaguchi University. M Furutani-Seiki was supported by the Takeda Science Foundation. We thank for technical support at the Yamaguchi University Science Research Center. The MR1 tetramer technology was developed jointly by Dr. James McCluskey, Dr. Jamie Rossjohn, and Dr. David Fairlie, and the material was produced by the NIH Tetramer Core Facility as permitted to be distributed by the University of Melbourne.

## Author Contributions

K Shibata: conceptualization, resources, data curation, software, formal analysis, supervision, funding acquisition, validation, investigation, visualization, methodology, project administration, and writing—original draft, review, and editing.
T Shimizu: resources, data curation, formal analysis, validation, investigation, and writing—review and editing.
M Nakahara: investigation.
E Ito: investigation.
F Legoux: resources, data curation, and writing—review and editing.
S Fujii: investigation.
Y Yamada: investigation.
M Furutani-Seiki: data curation.
O Lantz: resources, data curation, and writing—review and editing.
S Yamasaki: resources.
M Watarai: writing—review and editing.
M Shirai: funding acquisition and writing—review and editing.

## Conflict of Interest Statement

The authors declare that they have no conflict of interest.

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
