## [Reviewer comments · Life Science Alliance]

Life Science Alliance

The intracellular pathogen *Francisella* escapes from adaptive immunity by metabolic adaptation

Kensuke Shibata, Takashi Shimizu, Mashio Nakahara, Emi Ito, Francois Legoux, Shotaro Fujii, Yuka Yamada, Makoto Furutani-Seiki, Olivier Lantz, Sho Yamasaki, Masahisa Watarai, and Mutsunori Shirai

DOI: <https://doi.org/10.26508/lsa.202201441>

Corresponding author(s): *Kensuke Shibata, Yamaguchi University*

Review Timeline:

Submission Date:	2022-03-10
Editorial Decision:	2022-04-11
Revision Received:	2022-04-15
Editorial Decision:	2022-05-09
Revision Received:	2022-05-12
Accepted:	2022-05-13

Transaction Report:

April 11, 2022

Re: Life Science Alliance manuscript #LSA-2022-01441-T

Dr. Kensuke Shibata
Yamaguchi University
Department of Microbiology and Immunology
Minami-Kogushi, 1-1-1
Ube, Yamaguchi 755-8505
Japan

Dear Dr. Shibata,

Thank you for submitting your manuscript entitled "The intracellular pathogen *Francisella tularensis* escapes from adaptive immunity by metabolic adaptation" to Life Science Alliance. The manuscript was assessed by expert reviewers, whose comments are appended to this letter. We invite you to submit a revised manuscript addressing the Reviewer comments.

Thank you for this interesting contribution to Life Science Alliance. We are looking forward to receiving your revised manuscript.

Sincerely,

B. MANUSCRIPT ORGANIZATION AND FORMATTING:

Reviewer #1 (Comments to the Authors (Required)):

The authors propose that FT-type RibD is inefficient at the induction of human and murine MAIT cells. They provide compelling evidence that expression of FT RibD in FN leads to impaired expansion of MAIT cells and this is associated with increased susceptibility. This is an interesting and well written manuscript. The data is largely supportive of their claims and there arent any major issues with the data presented. Quantifying MAIT ligands from FN, FT, FN expressing FT RibD, and FN with delta RibD would further strengthen their arguments. Inclusion of representative flow data for NFAT GFP reporter should be added.

Reviewer #2 (Comments to the Authors (Required)):

Shibata and colleagues describe an instructive and relevant example of metabolic adaptation of an intracellular pathogen and characterize how disruption of bacterial riboflavin synthesis pathway in the i.c. pathogen mediates escape from recognition by MAIT cells.

Data are presented clearly and fully support the hypothesis. The MS is well-written.

Although it would of course be nice to show how and whether reversion of the riboflavin synthesis pathway in the pathogenic variant would impair pathogenicity in vivo, I understand that these experiments are difficult to execute and certainly beyond the scope of this MS.

Overall, I enjoyed reading the work and have only very minor remarks:

- page numbers are missing
- separation of results of results and discussion or at least a subheading "concluding remarks" at the start of the discussion part.

Reviewer #3 (Comments to the Authors (Required)):

Shibata, Shimizu and colleagues show convincingly that a pathogenic strain of *Francisella tularensis* has a metabolic adaptation in producing a mutant RibD protein, which reduces the production of riboflavin and hence avoids activation by MAIT cells. The authors use in vitro assays and in vivo mouse infections to show that introducing this mutation in ribD into a non-pathogenic strain avoid MAIT detection and can render it more lethal in mice.

There are some issues that need addressing as noted below, but over all this is a clear and concise manuscript that provides a valuable addition to the field of MAIT cell immunity and how pathogens can avoid this arm of immunity.

Major:

- Page 13: the authors suggest that the bacteria are visible in the cytoplasm, but this level of resolution does not show cytoplasmic location but rather the presence of FN in the lungs. Unless the analysis can be improved the description should just be modified, as it's not essential to this aspect to confirm they are cytosolic or not.
- Fig 4: Normally mouse replicates are combined from multiple experiments, rather than showing one representative experiment. As the numbers are small, particularly in H-J, the 3 replicate experiments should be shown. E.g. 3 naïve mice in H is not enough to draw solid conclusions from - if there is 2 other experiments performed these should be combined.
- Fig 4J: It is important to show naïve mice too. This will reveal if the number of DCs and macrophages expanded in FN, or rather contracted in FNΔribD infection.
- As the deletion of ribD in FN has no impact on growth, it strongly suggests that FN produces a riboflavin transporter to acquire exogenous riboflavin (this is the case for many pathogens such as *Listeria*, another intracellular pathogen lacking the riboflavin biosynthetic pathway). As FT bears mutations leading to non-functional RibD, the authors should address the presence of genes encoding such riboflavin transporter in FT (ribU-like, see well conserved sequences in other bacteria). Equally, as the authors have performed their RNAseq experiments using BHI, a rich nutrient containing riboflavin species, they could easily miss the changes in riboflavin transporter expression. If one is identified, the authors could use qPCR to show its expression in each

strain (FT, FN, WT and mutants) grown in minimal media (M63+/- riboflavin).

Minor:

- Page 4 "categorized as a Tier 1 Bioterrorism agent" Can you provide a reference for this (CDC)?
- Fig 2C: use some notation for the identical amino acids, most commonly asterisk; this aids in finding the mutations
- Brief description of what the antigen presenting cells are should be included in the results and legend - page 11. Are these MR1-over expressing cells?
- Fig 4I should ideally be before 4G - seems out of place there
- Fig 4 G-J: Gating strategy of MAIT cells should be included, as well as that for DCs and macrophages, at least in supplementary information, and the fluorophores should be shown on flow cytometry axes.
- The Results/discussion section is written as two parts, and if the journal's preference is to have combined it should be intertwined properly.
- Final paragraph: "we propose that a metabolite-targeted strategy, rather than traditional protein-targeted approaches" - can you elaborate how this would work? It is very different to protein antigens and thus should be briefly outlined how this is envisaged.
- Methods: EZ-Brads should be EZ-beads?
- Fig 3: what statistical test was used in E and F?
- Final point - it would be interesting to introduce the FN 'WT' RibD into the FT strain - to address the question of if this adaptation alone can attenuate the virulence. This is not essential to publication, and unlikely a feasible request in a reasonable timeframe, but could be posed as a question in the discussion.
- In the introduction: gene loss refers to genome reduction. As the authors are describing point mutations in ribD, they should also include in the introduction (first paragraph) and discussion that metabolic adaptation can occur by genetic changes leading to non-functional products.
- The overuse of the term "bacterial adaptation" across the text is not referenced properly. Please add comprehensive bioinformatic analyses supporting such claims

Reviewer #1 (Comments to the Authors (Required)):

The authors propose that FT-type RibD is inefficient at the induction of human and murine MAIT cells. They provide compelling evidence that expression of FT RibD in FN leads to impaired expansion of MAIT cells and this is associated with increased susceptibility. This is an interesting and well written manuscript. The data is largely supportive of their claims and there arent any major issues with the data presented. Quantifying MAIT ligands from FN, FT, FN expressing FT RibD, and FN with delta RibD would further strengthen their arguments. Inclusion of representative flow data for NFAT GFP reporter should be added.

I thank the reviewer for the comments. We totally agree with reviewer's suggestion that quantification of MAIT cell ligands would strengthen our arguments. To test this, we have been trying to identify the FN-specific molecules by HPLC analysis. We have identified some peaks detected only in FN homogenates but not in FN Δ ribD homogenates; however, after fractionation, we could not find any activity to MAIT reporter cell lines possibly due to unstable nature of MAIT cell ligands as previously described (Mak et al., 2017). Therefore, we have not included the immature data in the current manuscript.

In accordance with the reviewer's suggestion, we have added representative histograms in the revised Fig. S2.

Reviewer #2 (Comments to the Authors (Required)):

Shibata and colleagues describe an instructive and relevant example of metabolic adaptation of an intracellular pathogen and characterize how disruption of bacterial riboflavin synthesis pathway in the i.c. pathogen mediates escape from recognition by MAIT cells.

Data are presented clearly and fully support the hypothesis. The MS is well-written.

Although it would of course be nice to show how and whether reversion of the riboflavin synthesis pathway in the pathogenic variant would impair pathogenicity *in vivo*, I understand that these experiments are difficult to execute and certainly beyond the scope of this MS.

I thank the reviewer for the interesting suggestion; however, as mentioned by the reviewer, reversion of the riboflavin synthesis pathway in the pathogenic variants by introducing functional *ribD* to the pathogenic FT strain is difficult to test within this review period. Instead, we have further asked which amino acid substitutions in the FT strain are responsible for the pathogenesis *in vivo*. To address this question, we have done infection experiment using the FNribD^{FT(80)} strain, as its homogenate has the lowest MAIT cell agonistic activity in both human and mice (Fig. 3E, F). As shown below, we have not observed the change of its pathogenesis compared with the parental FN strain. This result suggests that other four substitutions are involved in the pathogenesis. As we are now carefully testing this possibility, we have not included the preliminary result in this manuscript.

Overall, I enjoyed reading the work and have only very minor remarks:

- page numbers are missing

- separation of results of results and discussion or at least a subheading "concluding remarks" at the start of the discussion part.

In accordance with the reviewer's suggestion, we have added page numbers and divided Results and Discussion section to two sections in the revised manuscript.

Reviewer #3 (Comments to the Authors (Required)):

Shibata, Shimizu and colleagues show convincingly that a pathogenic strain of *Francisella tularensis* has a metabolic adaptation in producing a mutant RibD protein, which reduces the production of riboflavin and hence avoids activation by MAIT cells. The authors use in vitro assays and in vivo mouse infections to show that introducing this mutation in *ribD* into a non-pathogenic strain avoid MAIT detection and can render it more lethal in mice.

There are some issues that need addressing as noted below, but over all this is a clear and concise manuscript that provides a valuable addition to the field of MAIT cell immunity and how pathogens can avoid this arm of immunity.

I thank the reviewer for constructive comments.

Major:

- Page 13: the authors suggest that the bacteria are visible in the cytoplasm, but this level of resolution does not show cytoplasmic location but rather the presence of FN in the lungs. Unless the analysis can be improved the description should just be modified, as it's not essential to this aspect to confirm they are cytosolic or not.

I apologize for inappropriate description. In accordance with the reviewer's suggestion, we have corrected the sentence not to specify the cytoplasmic localization in the revised manuscript (page 15, lines 4–7).

- Fig 4: Normally mouse replicates are combined from multiple experiments, rather than showing one representative experiment. As the numbers are small, particularly in H-J, the 3 replicate experiments should be shown. E.g. 3 naïve mice in H is not enough to draw solid conclusions from - if there is 2 other experiments performed these should be combined.

In accordance with the reviewer's suggestion, we have shown all data from multiple experiments in revised Fig. 4J (original Fig. 4H).

- Fig 4J: It is important to show naïve mice too. This will reveal if the number of DCs and macrophages expanded in FN, or rather contracted in FN Δ ribD infection.

In accordance with the reviewer's suggestion, we have added the data of naïve mice in the revised Fig. 4H (original Fig. 4 J) as shown below.

- As the deletion of ribD in FN has no impact on growth, it strongly suggests that FN produces a riboflavin transporter to acquire exogenous riboflavin (this is the case for many pathogens such as *Listeria*, another intracellular pathogen lacking the riboflavin biosynthetic pathway). As FT bears mutations leading to non-functional RibD, the authors should address the presence of genes encoding such riboflavin transporter in FT (ribU-like, see well conserved sequences in other bacteria). Equally, as the authors have performed their RNAseq experiments using BHI, a rich nutrient containing riboflavin species, they could easily miss the changes in riboflavin transporter expression. If one is identified, the authors could use qPCR to show its expression in each strain (FT, FN, WT and mutants) grown in minimal media (M63+/- riboflavin).

We thank the reviewer for pointing out an important issue to understand the pathogenesis of *Francisella* species. We have also been interested in whether gram-negative *Francisella* species can acquire exogenous riboflavin by its transporters. As suggested by the reviewer, we have looked for ribU-like genes related to riboflavin transporters in the Transporter Classification Database (<https://tcdb.org>). We found that, in bacteria, ribU-like genes belong to the Prokaryotic Riboflavin Transporter (P-RFT) Family; however, no P-RFT family members are shown to be identified in the Gram-negative bacterial subdivisions except Thermotogales (<https://tcdb.org/search/result.php?tc=2.A.87>). Nevertheless, we cannot exclude the possibility that unidentified riboflavin transporters are still present in *Francisella* species as recently found in other bacteria (Gutierrez-Preciado et al., 2015).

Another possible mechanism to support ribD-independent growth of *Francisella* species would be that the riboflavin synthetic pathway bypassing ribD compensate the defect (depicted in Fig. 2A). Supporting this idea, we could not generate a ribA-deficient FN strain even in the presence

of excessive amount of riboflavin. Therefore, we speculate that normal growth of the FN Δ ribD strain (Fig. 3A) may be dependent on ribA rather than riboflavin transporters in *Francisella* species.

Minor:

- Page 4 "categorized as a Tier 1 Bioterrorism agent" Can you provide a reference for this (CDC)?

Following the reviewer's comment, we have added the reference in the revised manuscript (page 4–5).

- Fig 2C: use some notation for the identical amino acids, most commonly asterisk; this aids in finding the mutations

In accordance with the reviewer's suggestion, we have indicated the amino acid substitutions by asterisk.

- Brief description of what the antigen presenting cells are should be included in the results and legend - page 11. Are these MR1-over expressing cells?

I apologize for unclear description in the Methods section of original manuscript. We have corrected the information in the revised manuscript (pages 21–22).

- Fig 4I should ideally be before 4G - seems out of place there

I agree with the reviewer's suggestion. Accordingly, we have changed the order and the corresponding sentences in the revised manuscript (pages 15–16).

- Fig 4 G-J: Gating strategy of MAIT cells should be included, as well as that for DCs and macrophages, at least in supplementary information, and the fluorophores should be shown on flow cytometry axes.

In accordance with the reviewer's suggestion, we have added gating strategy for MAIT cells, macrophages and DCs in Fig. S3 as shown below. We have also added the fluorophores in the revised Fig. 4 and Fig. S3.

- The Results/discussion section is written as two parts, and if the journal's preference is to have combined it should be intertwined properly.

Following the author guideline, we have separated the Results/discussion section into two parts in the revised manuscript.

- Final paragraph: "we propose that a metabolite-targeted strategy, rather than traditional protein-targeted approaches" - can you elaborate how this would work? It is very different to protein antigens and thus should be briefly outlined how this is envisaged.

In accordance with the reviewer's suggestion, we have added the sentences describing the difference between protein-targeted strategy and metabolite-targeted strategy in the revised manuscript (page 20, lines 4–13).

- Methods: EZ-Brads should be EZ-beads?

We apologize for the mistake. We have corrected the error in the revised manuscript.

- Fig 3: what statistical test was used in E and F?

We apologize for the lack of the description. We have added the description in the revised manuscript (page 41, lines 2–4).

- Final point - it would be interesting to introduce the FN 'WT' RibD into the FT strain - to address the question of if this adaptation alone can attenuate the virulence. This is not essential to publication, and unlikely a feasible request in a reasonable timeframe, but could be posed as a question in the discussion.

I thank the reviewer for an interesting suggestion; however, as mentioned by the reviewer, we require additional application for genetic manipulation of the pathogenic strain. As we cannot test this within this review period, we have added the description for the future study in the revised manuscript (page 17, lines 8–9).

- In the introduction: gene loss refers to genome reduction. As the authors are describing point mutations in ribD, they should also include in the introduction (first paragraph) and discussion that metabolic adaptation can occur by genetic changes leading to non-functional products.

I agree with the reviewer's suggestion. Accordingly, we have changed the sentence describing that, in addition to gene loss, genome reduction can be induced by genetic changes in the revised manuscript (page 4, lines 4–5 / page 18, lines 4–7).

- The overuse of the term "bacterial adaptation" across the text is not referenced properly. Please add comprehensive bioinformatic analyses supporting such claims.

I apologize for the unclear description of the term "bacterial adaptation". As some gene expression profiles observed in the FT strain in Fig. 1 are previously shown to be involved in bacterial adaptation, we referred to this phenomenon as "metabolic adaptation". Accordingly, we have carefully added this description in the Results section of the revised manuscript (page 10–11).

References

- Gutierrez-Preciado, A., A.G. Torres, E. Merino, H.R. Bonomi, F.A. Goldbaum, and V.A. Garcia-Angulo. 2015. Extensive Identification of Bacterial Riboflavin Transporters and Their Distribution across Bacterial Species. *PLoS One* 10:e0126124.
- Mak, J.Y., W. Xu, R.C. Reid, A.J. Corbett, B.S. Meehan, H. Wang, Z. Chen, J. Rossjohn, J. McCluskey, L. Liu, and D.P. Fairlie. 2017. Stabilizing short-lived Schiff base derivatives of 5-aminouracils that activate mucosal-associated invariant T cells. *Nat Commun* 8:14599.

May 9, 2022

RE: Life Science Alliance Manuscript #LSA-2022-01441-TR

Kensuke Shibata

Dear Dr. Shibata,

Thank you for submitting your revised manuscript entitled "The intracellular pathogen *Francisella* escapes from adaptive immunity by metabolic adaptation". We would be happy to publish your paper in Life Science Alliance pending final revisions necessary to meet our formatting guidelines.

- please add a Running Title to our system
- please add the Twitter handle of your host institute/organization as well as your own or/and one of the authors in our system
- please use the [10 author names, et al.] format in your references (i.e. limit the author names to the first 10)
- please add your supplementary figure legends and your table legend to the main manuscript text, directly under the main figure legends
- we encourage you to introduce the panels in the figure legends in alphabetical order
- please provide your Table file in editable doc or excel format or include it in the doc file of your manuscript text

To upload the final version of your manuscript, please log in to your account: <https://lsa.msubmit.net/cgi-bin/main.plex>. You will be guided to complete the submission of your revised manuscript and to fill in all necessary information. Please get in touch in case you do not know or remember your login name.

A. FINAL FILES:

B. MANUSCRIPT ORGANIZATION AND FORMATTING:

****It is Life Science Alliance policy that if requested, original data images must be made available to the editors. Failure to provide**

original images upon request will result in unavoidable delays in publication. Please ensure that you have access to all original data images prior to final submission.**

The license to publish form must be signed before your manuscript can be sent to production. A link to the electronic license to publish form will be sent to the corresponding author only. Please take a moment to check your funder requirements.

Sincerely,

Reviewer #1 (Comments to the Authors (Required)):

I am satisfied with the authors comments on addressing my concerns.

Reviewer #2 (Comments to the Authors (Required)):

The authors have addressed the remaining minor concerns of the three reviewers. It's a great story for the MAIT cell community and beyond.

Reviewer #3 (Comments to the Authors (Required)):

The authors have adequately addressed all of my original concerns. The manuscript will be a valuable addition to the field.

May 13, 2022

RE: Life Science Alliance Manuscript #LSA-2022-01441-TRR

Dr. Kensuke Shibata
Yamaguchi University
Microbiology and Immunology
Minamikogushi, 1-1-1
Ube 755-8505
Japan

Dear Dr. Shibata,

Thank you for submitting your Research Article entitled "The intracellular pathogen *Francisella* escapes from adaptive immunity by metabolic adaptation". It is a pleasure to let you know that your manuscript is now accepted for publication in Life Science Alliance. Congratulations on this interesting work.

DISTRIBUTION OF MATERIALS:

Again, congratulations on a very nice paper. I hope you found the review process to be constructive and are pleased with how the manuscript was handled editorially. We look forward to future exciting submissions from your lab.

Sincerely,
